# Analytical Separation of Closantel Enantiomers by HPLC

**DOI:** 10.3390/molecules26237288

**Published:** 2021-11-30

**Authors:** Basma Saleh, Tongyan Ding, Yuwei Wang, Xiantong Zheng, Rong Liu, Limin He

**Affiliations:** 1National Reference Laboratory of Veterinary Drug Residues (SCAU), College of Veterinary Medicine, South China Agricultural University, Guangzhou 510642, China; basmasaleh30@gmail.com (B.S.); dingtongyan1994@163.com (T.D.); 13414971931@163.com (X.Z.); 2Directorate of Veterinary Medicine, General Organization of Veterinary Services, Ministry of Agriculture, Port Said 42511, Egypt; 3Quality Supervision, Inspection and Testing Center for Domestic Animal Products Guangzhou, Ministry of Agriculture and Rural Affairs, Guangzhou 510642, China; 18819427447@163.com (Y.W.); lljrr@scau.edu.cn (R.L.)

**Keywords:** chiral stationary phase, closantel, enantiomeric separation, high-performance liquid chromatography, modifiers

## Abstract

Closantel is an antiparasitic drug marketed in a racemic form with one chiral center. It is meaningful to develop a method for separating and analyzing the closantel enantiomers. In this work, two enantiomeric separation methods of closantel were explored by normal-phase high-performance liquid chromatography. The influences of the chiral stationary phase (CSP) structure, the mobile phase composition, the nature and proportion of different mobile phase modifiers (alcohols and acids), and the column temperature on the enantiomeric separation of closantel were investigated in detail. The two enantiomers were successfully separated on the novel CSP of isopropyl derivatives of cyclofructan 6 and *n*-hexane-isopropanol-trifluoroacetic acid (97:3:0.1, *v*/*v*/*v*) as a mobile phase with a resolution (*R*s) of about 2.48. The enantiomers were also well separated on the CSP of *tris*-carbamates of amylose with a higher *R*s (about 3.79) when a mixture of *n*-hexane-isopropanol-trifluoroacetic acid (55:45:0.1, *v*/*v*/*v*) was used as mobile phase. Thus, the proposed separation methods can facilitate molecular pharmacological and biological research on closantel and its enantiomers.

## 1. Introduction

Chiral recognition occurs mainly in natural and chemical systems, and this phenomenon has had a fundamental impact in various fields [1]. Analytical chemists have considered chiral separation, stereochemistry discovery, and enantioselective studies of chiral compounds as major tasks in modern academic research. In the last few decades, the increase in the demand for producing pure single enantiomer drugs has been driven by the potential advantages concerning safety and efficiency [2,3,4], because it is well-known that the therapeutic effect of a racemic drug is usually produced by one enantiomer, while the other might be either inactive or have a toxic or synergetic effect [5]. Therefore, there has been a widespread interest in chiral drugs separation and analysis in the pharmaceutical industry. In addition, numerous scientific research has focused on developing effective chromatographic methods for separating the racemic drugs [6,7,8,9]

High-performance liquid chromatography (HPLC) is an ideal tool for chiral drugs separation and analysis due to its inherent accuracy, specificity, and versatility. The optimization process of the chromatographic condition is time-consuming, owing to the influences of many factors such as the composition of the mobile phase [10], structure of the chiral stationary phase (CSP) [11], and the column temperature [12]. It is necessary to explore the influence of these factors on chiral recognition. In recent years, numerous CSPs have been modified with various chiral selectors for effective chiral separations. Not only the proper choice of CSP but also the optimization of mobile phase modifiers are needed to achieve higher enantioselectivity, which is realized through improving the complementary interactions between the analyte and functional groups on the chiral selector [13].

Closantel is an anti-parasitic drug, a synthetic halogenated salicylanilide derivative compound, namely N-[5-chloro-4-[(4-chlorophenyl)(cyano)methyl]-2-methylphenyl]-2-hydroxy-3,5-diiodobenzamide. As a typical chiral compound, the closantel racemic (*rac*) mixture contains an SP^3^-hybridized carbon atom that carries four substituents and exists as a pair of enantiomers, as shown in Appendix A.

To date, few studies have investigated the chiral separation of closantel enantiomers using HPLC [14]. This research aimed to develop analytical separations of closantel enantiomers using the normal-phase (NP) HPLC mode with different types of chiral selectors such as *tris* carbamates of amylose polysaccharides [15,16], isopropyl carbamates of cyclofructans [17,18,19], and β-cyclodextrins derivatives [20]. Filling the gap in the analytical separation of racemic drugs has great significance for the chemical, pharmaceutical, and biological fields. Hence, this work would provide effective support and reference for any further research related to closantel enantiomers. In addition, the effects of CSP structure, mobile phase modifiers, and column temperature on the enantiomeric separation of closantel are discussed in detail.

## 2. Materials and Methods

### 2.1. Chemicals and Materials

Standard *rac*-closantel was purchased from Dr. Ehrenstorfer GmbH Company (Augsburg, Germany) (>98.1%, purity). Pure closantel enantiomers were provided from Guangzhou Yan Chuang Biotechnology Development Co., Ltd. (Guangzhou, China) (>98%, purity) for the first and second enantiomers. Isopropanol (IPA), ethanol (EtOH), methanol (MeOH), acetonitrile (ACN), and *n*-hexane of HPLC grade were purchased from Fisher Scientific Company (Fairlawn, NJ, USA). Trifluoroacetic acid (TFA), formic acid (FA), acetic acid (AcOH), and triethylamine (TEA) were purchased from Sigma Aldrich Company, (Hamburg, Germany). Four different analytical chiral columns were used and are listed in Table 1: Enantiopak^®^ SCDP [21], Enantiopak^®^Y3R [22], Chiralpak^®^AD−3 [23], and Poroshell 120 CF6 [24]. The four chiral columns have the same internal diameter (4.6 mm i.d.) and were loaded with different particle sizes of silica matrices.

### 2.2. Chromatographic Conditions

The analytical separation of closantel was carried out using the Waters Alliance 2695 Separations Module HPLC system including a 600E controller pump, 776 auto-sampler, and 2487 dual-wavelength ultraviolet (UV) detector (Waters Corporation, Milford, CT, USA). The ultrasonic cleaning system K Q5200B type (Kunshan city ultrasonic instrument Co., Ltd., Kunshan, China) and micropipette gun Pipetman type (Gillson Company, Middleton, WI, USA) were used. Empower 3 software combined the data from UV peaks to give the peak information of each enantiomer and was also used for the system control. Deadtime (*t*_0_) was estimated by the first disturbance in the baseline at a 1 mL/min flow rate for the four columns unless mentioned. The dead times for Chiralpak AD−3, Enantiopak SCDP, and Enantiopak Y3-R columns were all 3.0 min, whereas *t*_0_ was 1.5 min for Poroshell 120 CF6, due to the differences in length.

### 2.3. Sample Preparation

Amounts of 1 mg/mL stock solutions were prepared by accurately weighing about 10 mg of *rac*-closantel and its pure enantiomers separately in a 10 mL volumetric flask, then accurately adding IPA to reach the volume scale. The stock solutions were stored in a 10 mL centrifuge tube covered with aluminum foil in the dark at −15 °C. The injection volume was 5 µL for Infinity Lab Poroshell 120 CF6, Enantiopak^®^ SCDP, Enantiopak^®^ Y3-R, and Chiralpak AD−3 columns in triplicate.

### 2.4. Optimization of Chromatographic Conditions

Four analytical columns with various chiral selectors were selected and screened using the NP-HPLC separation mode. The mobile phase was a mixture of an alkane (*n*-hexane) with different proportions of an alcohol modifier: EtOH, IPA, or MeOH. In addition, the acidic modifier was TFA, AcOH, or FA. The flow rate was 1 mL/min unless mentioned, the column temperature was 25 °C unless mentioned, and the detection wavelength was 231 nm.

The retention factors (*k*) were determined with the following equation:(1)$$k=\frac{\left(t_{R}-t_{0}\right)}{t_{0}}\;\;$$
where ($t_{R}$) is the retention time for the first or second enantiomers and $t_{0}$ is the dead time.

The separation factor or selectivity (*α*) was determined by the following equation:(2)$$\alpha =\frac{k_{2}}{k_{1}}$$
where *k*_1_ and *k*_2_ are the retention factors for the first and second enantiomers, respectively.

Resolution factor (*R*s) was determined by the following formula:(3)$$Rs=\frac{1.18\;\left(t_{R}{}_{2}-t_{R}{}_{1}\right)\;}{\left(w_{0.5_{1}+}w_{0.5_{2}}\right)}$$
where $w_{0.5_{1}}$ and $w_{0.5_{2}}$ are the peak widths at half the peak height for the first and second enantiomers, respectively.

Theoretical plate number (*N*) was determined by the following formula:(4)$$N=5.54{(\;\frac{t_{R}}{w_{0.5}})}^{2}$$

### 2.5. Racemization and Transformation of Closantel Enantiomers

The influences of acidic (TFA or FA) and basic additives (TEA), storage time (0–72 h), and temperature (−20 °C, 4 °C, and 25 °C) on the racemization of closantel enantiomers in different organic solvents (IPA, acetone, MeOH, and ACN) were studied in detail on a Chiralpak AD−3 column in normal-phase mode.

## 3. Results and Discussion

In this work, a direct HPLC separation method of closantel enantiomers was developed. In the following results, the factors affecting the chiral separation of closantel enantiomers are described in detail.

### 3.1. Effect of CSP Structure on the Enantiomeric Separation of Closantel

It is well-known that the selectivity of CSP to an enantiomer comes from the specific interactions between the enantiomer and the stereo-specific sites on the chiral selector. Therefore, it is necessary to explore the influence of the different structures of CSPs throughout the separation of *rac*-closantel. As mentioned in the Methods section, four different CSPs: Chiralpak^®^ AD−3, Infinity Lab Poroshell 120^®^ CF6, Enantiopak^®^ SCDP, and Y3R, were scrutinized under normal-phase conditions, using *n*-hexane combined with different proportions of alcohol modifiers (IPA, EtOH, or MeOH) and acidic modifiers (TFA, AcOH, or FA). Based on the results, it was observed that the enantiomeric separation of closantel was completely different on the four columns used, as shown in Figure 1. For instance, under the optimal conditions using a mobile phase consisting of 0.1% TFA in the mixture of *n*-hexane-IPA with appropriate proportions, only one chromatographic peak was obtained by the Enantiopak Y3-R column (Figure 1A). The separation was partial on the Enantiopak SCDP column (*R*s = 0.98), as shown in Figure 1B. However, the baseline separation of the two enantiomers was successfully achieved on Poroshell 120 CF6 using 3% IPA in the mobile phase, where *R*s was 2.48 within 15 min (Figure 1C), and on Chiralpak AD−3 using 45% IPA, where *R*s was 3.79 within 6 min (Figure 1D).

There are several possible explanations for these results; first, Chiralpak^®^ AD−3 is a derivatized amylose CSP, which consists of a stereo-regular sequence of *D-glucopyranose* units arranged along the axis forming helical grooves. Each unit contains five chiral centers, and the closantel studied contains one chiral center. Therefore, the supramolecular structure of the amylose backbone and the derivatized groups contribute to many potential interaction sites that have great remarkable properties in terms of enantioselectivity [23]. The intermolecular interactions that take place between the carbamate moiety and the analyte can affect the enantioselectivity. Moreover, the position of the substituents fused with the phenyl moiety also had a significant influence on chiral recognition. Therefore, the stereo-electronic, dipole–dipole, π–π bonding, and hydrogen bonding interactions can form evanescent, transient diastereomer complexes through interactive forces, thus leading to differences in the migration between the enantiomers inside the column causing enantioseparation [25]. The ADMPC (amylose *tris* 3, 5 dimethyl phenyl carbamates) enantioselectivity results not only from *D-glucopyranose* residues of amylose but also from helical grooves existing in the polymeric backbone, of which the difference in their molecular environment is considered to have a strong impact on the chiral recognition [26].

Secondly, Infinity Lab Poroshell 120 CF6 is a novel CSP of isopropyl carbamates cyclofructans 6 (IP-CF6), consisting of a natural crown ether solid core, where 6 *D-fructofuranose* units are oriented alternatively around the center with hydroxy groups [24,27] and supported onto the outer layer of superficially porous particles (SPPs) technology with 2.7 µm-size particles that provides high enantioselectivity with shorter retention times [20,28,29]. Each *D-fructofuranose* unit contains four chiral centers and three hydroxyl groups that provide hydrophilic properties of the chiral selectors [30]. The substituents attached with the derivatized group can disrupt the internal hydrogen bonding of CF6, causing relaxation of the molecular structure that exposes the crown ether core and hydrogen bonding in the polymer. Therefore, it can increase the steric bulk, which leads to an increase in the separation capability and improves peak efficiency.

Thirdly, Enantiopak SCDP is a modified β-cyclodextrin bound with silica gel, consisting of cyclic oligosaccharides in a doughnut shape, comprising seven α-(1,4)-glycosidic linkages, and fourteen hydroxyl groups positioned in the inner cavity. The presence of hydroxy groups allows several potential interactions with the analyte that lead to the enantiomeric separation. Enantiopak SCDP is also derivatized by chlorophenyl carbamate moieties with single urea linkages that can modulate the enantiomeric separation by providing multiple interactions such as π–π, dipole–dipole interaction, electrostatic interaction, and hydrogen bonding, which contributes significantly to effective enantiomeric separation. Nevertheless, Enantiopak SCDP has no separation capability for closantel enantiomers with all types and proportions of alcohol modifiers used, due to the possibilities of disruption of the internal hydrogen bonds, or the blocking of the inner cavity of cyclodextrin by the normal mobile phase components, as mentioned in previous studies [7,31]. The analyte might only interact with hydroxyl groups of cyclodextrin.

Fourthly, Enantiopak Y3-R is a modified amylose in a silicone surface, derivatized in the form of [(S)-α-methyl phenyl carbamate], has the ability of hydrogen bonding, and has some hydrophobic interactions. The modified amylose column (Chiralpak AD−3) showed the best performance in separating the closantel enantiomers. However, Enantiopak Y3-R has no separation capability for the closantel enantiomers, which is probably due to the difference in nature and position of the respective substituents on the aromatic moieties. In addition, the suppliers of the two modified amylose polymeric backbones are also different. Table 2 demonstrates the chromatographic separation parameters of closantel enantiomers on different types of chiral columns under various normal-phase conditions.

### 3.2. Optimization of Mobile Phase Compositions

Herein, Chiralpak AD−3 and Poroshell 120 CF6 columns were further investigated for closantel enantiomeric separation as they showed acceptable resolutions in Table 2.

#### 3.2.1. The Effect of the Nature and Proportion of Alcohol Modifiers on the Enantiomeric Separation of Closantel

A mixture of alkanes and different alcohol modifiers is usually applied as a mobile phase owing to their versatility in providing enantiomeric separations, due to the fact that hydrogen bonding, dipole–dipole, and π–π interactions, essential for chiral recognition, are more effective under normal-phase conditions. Hence, the normal mobile phase mode was selected to be used with the previously mentioned CSPs. The effect of three alcohol modifiers (IPA, EtOH, or MeOH) was assessed on the enantiomeric separation of closantel by changing the ratios of *n*-hexane: alcohol in the range from (75:25, *v*/*v*) to (55:45, *v*/*v*) on Chiralpak AD−3, and in the range from (97:3, *v*/*v*) to (80:20, *v*/*v*) on Poroshell 120 CF6. It was observed that the nature and proportion of the mobile phase could substantially change the chiral recognition on Chiralpak AD−3 and Poroshell 120 CF6. Not only the type of alcohol modifier but also its proportion in the mobile phase had a significant effect on the selectivity and retention times of the enantiomers. As shown in Figure 2A, the *R*s values showed an upward trend on Chiralpak AD−3 when the alcohol proportion increased. The other chromatographic parameters increased as well, with a significant decrease in the retention times, as illustrated in Figure 2C. When the proportion of IPA increased from 25% to 55%, the *α*, *R*s, *N*_1_, and *N*_2_ values improved from (2.58, 3.11, 973, and 760) to (2.81, 3.79, 3330, and 1845), respectively, with a marked decrease in the retention times. In contrast, the *R*s values obtained by Poroshell 120 CF6 showed a downward trend when the alcohol proportion increased, as shown in Figure 2B. The *α*, *R*s, *N*_1_, and *N*_2_ values decreased from (1.20, 2.48, 9773, and 4086) to (1.10, 1.18, 7852, and 7705) with shorter retention times, as illustrated in Figure 2D, when the proportion of IPA increased from 3% to 15%. In consideration of the results obtained with the three alcohol modifiers, it was evident that the application of different alcohol modifiers produced a significant variation in the resolution on Chiralpak AD−3 and Poroshell 120 CF6. The superiority of IPA over the other two alcohols was apparent. The highest potency performance of IPA in separation was proved through the values of resolution and selectivity, where (*α* = 2.81 and *R*s = 3.79) on Chiralpak AD−3 when the enantiomeric separation was enhanced by 45% IPA. In contrast, the separation profiles enhanced by 45% MeOH or 45% EtOH were lower, where (*α* = 2.00 and *R*s = 2.38) for MeOH and (*α* = 1.50 and *R*s = 1.43) for EtOH. Similarly, on Poroshell 120 CF6, (*α* = 1.20 and *R*s = 2.48) were obtained when 3% IPA was used as an alcohol modifier. However, the separation profile enhanced by 3% EtOH was relatively poor (*α* = 1.50 and *R*s = 1.56). When the mobile phase containing MeOH was used, no separation of closantel enantiomers was achieved. As shown from the results presented in Table 2 and Figure 2, the enantiomeric resolution and selectivity were improved by increasing the alcohol proportion on Chiralpak AD−3. On the contrary, the decrease in the alcohol proportion on Poroshell 120 CF6 improved significantly the enantiomeric resolution and selectivity.

The above results indicated that the branched alcohol (IPA) has a higher ability to modify the steric environment in chiral grooves of the amylose polymeric backbone, which likely seems to be correlated with the high steric size of IPA. Thus, the bulkiness of IPA could alter the size and shape of the helical grooves to become more favorable for enantioseparation [18]. Therefore, the reason for the enantioselectivity on Chiralpak AD−3 strongly relies on the modification of chiral grooves that lets portions of the closantel enantiomers enter and exhibit interactions with the active part of the polymer. Further, the mobile phase containing MeOH had a negative effect on the efficiency of the Chiralpak AD−3 column, and the retention time and resolution obtained from MeOH were also almost constant. There are explanations for these findings: MeOH is a protic solvent, which can strongly be engaged with the chiral selector due to its high polarity that contributes to breaking the hydrogen bonding interaction between the analyte and ADMPC polymer [32]. For this reason, the MeOH could reduce the chiral recognition and separation [33,34], in contrast with EtOH and IPA that have lower polarity. In addition to the poor adsorption of MeOH in the chiral polymeric structure, the distribution peak of MeOH in the ADMPC backbone was much lower than those of EtOH and IPA, as mentioned in [35]. On Poroshell 120 CF6, the typical normal-phase behavior was observed, in which the alcohol modifier affected the enantioselectivity through the steric hindrance rather than hydrogen bonding and dipole–dipole interactions. As a result of the steric hindrance, the *R*s values increased with longer retention times when IPA was decreased in the mobile phase. As mentioned in a previous study, the mobile phase containing IPA (long-chain alcohol) achieved better selectivity than EtOH in enantiomeric separation on the modified isopropyl carbamates of cyclofructan 6 chiral selector [36]. As MeOH is a short-chain alcohol, this is probably the reason for no separation of closantel enantiomers when the mobile phase containing MeOH was used. It was obvious from the experimental observation that the Poroshell 120 CF6 column exhibited higher theoretical plate number (*N*) values, owing to its smaller particle size and the improvement in particle technology (SPP) [37]. Moreover, the enantiomers elution order (EEO) of the closantel enantiomers was constant without interconversion on Chiralpak AD−3 and Poroshell CF6 under all developed conditions.

#### 3.2.2. Effect of Acidic Modifiers on the Enantiomeric Separation of Closantel

As mentioned in previous studies [21,22,23,24], a minor proportion of an acidic modifier is sometimes added to the eluent, particularly during the chiral separation of acidic analytes for efficient enantioselective separation. Herein, more details on the effect of acidic modifiers would be given. The effect of three acidic modifiers (TFA, AcOH, or FA) has been evaluated on the enantiomeric separation of closantel. The results showed that TFA was more efficient than FA and AcOH as an acidic modifier for separating the closantel enantiomers owing to its high acidity strength, where the values of *α* and *R*s were highest. However, FA had the worst effect on enantiomeric separation, where the values of *α* and *R*s were lowest and the peaks were too wide, peak tailing was serious, and column efficiency was also reduced. Table 3 shows the comparison of the effect of the three acidic modifiers on the enantiomeric separation of closantel on Chiralpak AD−3 and Poroshell 120 CF6, respectively. Under the optimal normal-phase conditions mentioned above, appropriate proportions of TFA were applied by variation in the range of (0.05–0.25%). On the other hand, AcOH or FA was applied in the range of (0.10–0.50%).

The results in Table 3 revealed that the acidic additive has a significant impact on the separation of closantel enantiomers. As shown also in Figure 3A, the *R*s values achieved on Chiralpak AD−3 improved from 2.80 to 3.28 and from 2.48 to 2.99 when the proportion of AcOH or FA increased from 0.10% to 0.50%, respectively. On the other hand, the *R*s values were approximately similar (*R*s = 3.75) without showing any trend when the proportion of TFA was changed from 0.05% to 0.25% (Figure 3A). The *R*s values achieved on Poroshell 120 CF6 first increased from 1.88 to 2.69 when the proportion of TFA increased from 0.05% to 0.1%. Although the *R*s values were negatively affected by increasing the proportion of TFA than 0.1%, the *R*s values unexpectedly decreased (*R*s = 1.04) when 0.5% TFA was added to the mobile phase with an unstable baseline. In addition, when the AcOH or FA proportion increased from 0.10% to 0.25% in the mobile phase, the *R*s values increased from 1.18 to 1.89 and from 1.35 to 1.86 with AcOH and FA, respectively. However, a slight decrease in *R*s values was shown when the AcOH or FA proportion added to the mobile phase was 0.50%, where the *R*s values were 1.80 and 1.58 for AcOH and FA, respectively (Figure 3B). For this reason, the slightly acidic mobile phase was found to be the suitable choice for the enantiomeric separation of closantel in Poroshell 120 CF6. In addition, the high acidity modifier proportion may be harmful to the HPLC system. Therefore, the 0.1% TFA in the mobile phase was chosen for further experiments. From Table 3 and Figure 3, the results are consistent with the published studies [38,39], which proved that the acidic additives were able to adjust the CSP surface charge by minimizing the deleterious effect of the residual free silanols on the silica surface, and they can also avoid the analyte ionization that enhances the interaction between the mobile phase and the analyte. In addition, the acidic modifiers can interact with the basic sites of the chiral selector to improve the peak shape and avoid asymmetry by masking nonenantioselective retention sites on CSP [40,41].

### 3.3. Influence of Column Temperature

The influence of the column temperature on the enantiomeric separation was investigated. Under normal-phase conditions, the effect of different column temperatures was discussed using the mobile phase ratio of *n*-hexane-IPA-TFA (97:3:0.1, *v*/*v*/*v*) on Poroshell 120 CF6. The experimental results revealed that the retention times and resolutions increased with a gradual decrease in temperature from 35 °C to 20 °C at a steady interval of 5 °C (from I to IV), as illustrated in (Figure 4).

As it can be observed from the results in Figure 4, the best enantiomeric separation was achieved on Poroshell 120 CF6, using *n*-hexane-IPA-TFA (97:3:0.1, *v*/*v*/*v*) at 20 °C, where the *α* and *R*s values improved from (1.20 and 2.48) to (1.25 and 2.77), respectively. In addition, the retention times increased from (10.5 and 11.9 min) to (11.6 and 13.4 min), respectively. As mentioned in previous studies [42,43], the lower temperature of the column resulted in higher resolutions, longer retention times, and wider peaks. Similarly, the lower temperature of Poroshell 120 CF6 showed a clear advantage on the enantiomeric separation of closantel. However, there was no significant change in the enantiomeric separation when the column temperature was changed on Chiralpak AD−3 and Enantiopak SCDP.

Finally, the best enantiomeric separation was obtained when the mobile phase was a mixture of *n*-hexane-IPA-TFA (55:45:0.1, *v*/*v*/*v*) at 25 °C in Chiralpak AD−3. However, the low-polarity mobile phase consisting of *n*-hexane-IPA-TFA (97:3:0.1, *v*/*v*/*v*) was the proper chromatographic condition used for the enantiomeric separation on Poroshell 120 CF6, as shown in Figure 5. Then, the above mobile phase compositions were designated for further experiments.

### 3.4. Racemization of Closantel Enantiomers 

As demonstrated in previous studies, the racemization of the pure enantiomers might occur under high temperature, organic solvent, or acidic/basic conditions [44,45]. Therefore, it is necessary to evaluate the racemization of the enantiomers. Through the experimental observations, the influence of pure organic solvents such as MeOH, ACN, IPA, or acetone on the racemization of closantel enantiomers could be neglected in low temperatures (below 10 °C) for one month.

#### 3.4.1. Influence of Acidic or Basic Additives

Individual enantiomer (1 mg/mL) stock solution was prepared by dissolving in MeOH with 0.1% TFA, FA, or TEA. Under the appropriate conditions, using *n*-hexane-IPA-TFA (55:45:0.1, *v*/*v*/*v*) as a mobile phase on Chiralpak AD−3, the concentrations of each enantiomer at different times were determined. The enantiomeric excess (*ee*) values were the peak area ratio between the first and second enantiomers, which remained stable with acidic additive (0.1% TFA or 0.1% FA) for 72 h. Therefore, the addition of 0.1% TFA or 0.1% FA to the solution of the two enantiomers could successfully inhibit the racemization. However, they were unstable in the basic additive solution (0.1% TEA). The racemization was fast for closantel enantiomers in the solution system containing 0.1% TEA, and the peak area ratio between two enantiomers showed a significant downward trend from 1 h to 12 h, as shown in (Figure 6A,B). Obviously, the basic additive was a critical factor affecting the racemization of closantel, as it could accelerate the racemization of the pure enantiomers.

#### 3.4.2. Influence of Storage Temperature

The effect of temperature on racemization was also investigated. The results indicated that the *ee* values for the enantiomers remained stable for 14 days at different storage temperatures of −20 °C, 4 °C, and 25 °C.

## 4. Conclusions

In summary, the rapid, precise, and efficient chiral separation methods for closantel enantiomers were established. The chiral stationary phases based on isopropyl carbamates cyclofructan 6 and *tris* carbamates of amylose were screened for the enantiomeric separation of closantel using normal-phase high-performance liquid chromatography, and the resolutions on the two columns were more than 2.5. Isopropanol proved to be the most suitable alcohol modifier in the mobile phase. This work contributes to better knowledge about the enantiomeric separation and chiral recognition of *rac*-closantel, and the proposed methods can facilitate molecular pharmacology on closantel enantiomers.

## Figures and Tables

**Figure 1 molecules-26-07288-f001:**
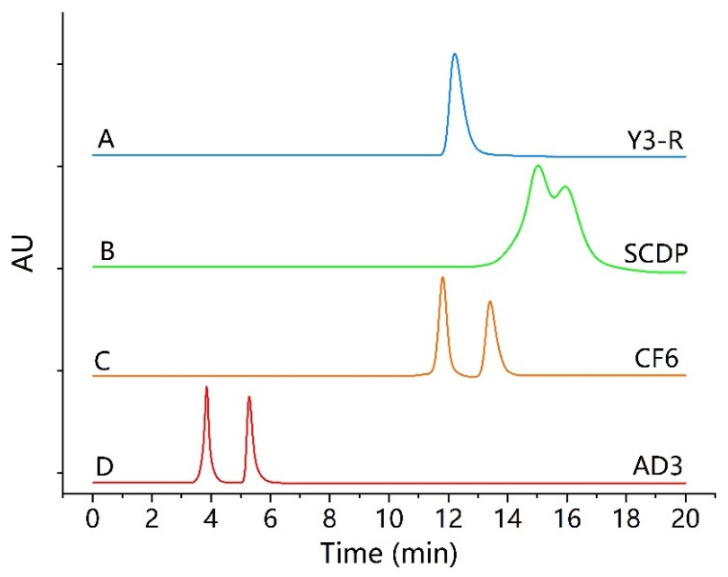
Influence of different CSP structures on the enantioseparation of closantel. (**A**) EnantiopakY3-R, (**B**) Enantiopak SCDP, (**C**) Poroshell 120 CF6, and (**D**) Chiralpak AD−3. Mobile phase, n-hexane-IPA-TFA with different ratios: for (**A**,**B**) (50:50:0.1, *v*/*v*/*v*), for (**C**) (97:3:0.1, *v*/*v*/*v*), and for (**D**) (55:45:0.1, *v*/*v*/*v*), at 1 mL/min flow rate, 5 µL injection volume from 1 mg/mL of closantel; UV-detection, 231 nm; at room temperature (25 °C), except for (**C**) Poroshell 120 CF6 (flow rate, 0.5 mL/min, at 20 °C column temperature).

**Figure 2 molecules-26-07288-f002:**
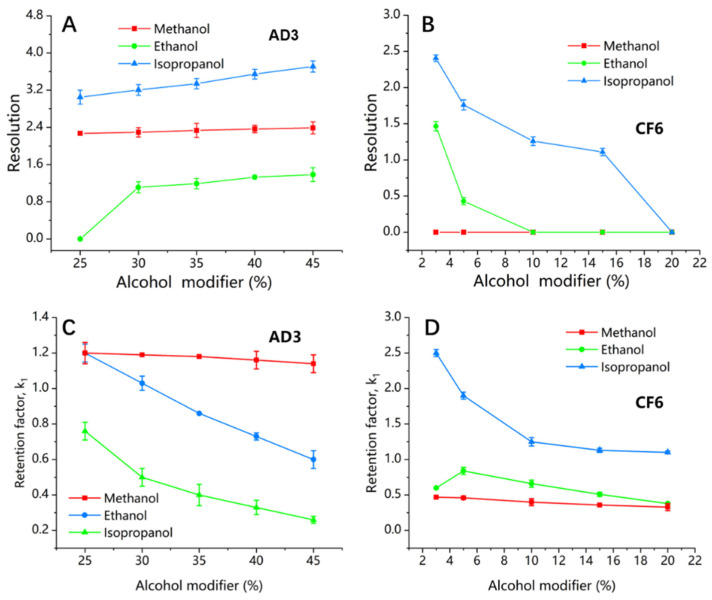
Influence of the alcoholic modifiers on the resolutions and retention times of closantel first eluting enantiomer on Chiralpak AD3 (**A**,**C**) and Poroshell 120 CF6 (**B**,**D**), respectively. k1, the retention time of first closantel enantiomer. Mobile phase, n-hexane-alcohol modifier −0.1% TFA; on Chiralpak AD3 (**A**,**C**) flow rate, 1 mL/min, and column temperature, 25 °C; on Poroshell 120 CF6 (**B**,**D**) flow rate, 0.5 mL/min, and column temperature, 20 °C.

**Figure 3 molecules-26-07288-f003:**
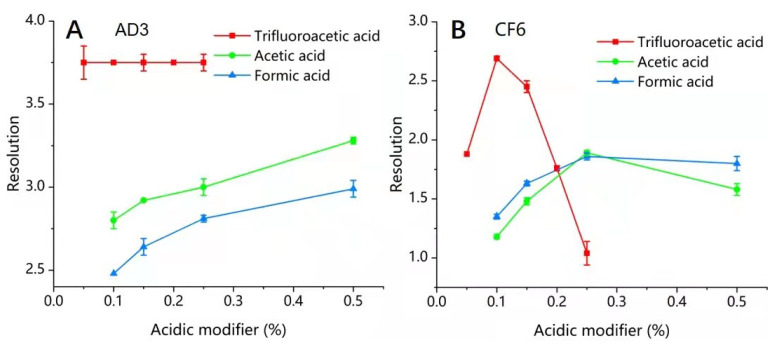
Influence of the acidic modifiers on the resolution of closantel enantiomers. (**A**) Mobile phase, *n*-hexane-IPA (55:45, *v*/*v*) with different proportions of acidic modifier; flow rate, 1 mL/min; column temperature, 25 °C; and (**B**) mobile phase, *n*-hexane-IPA (97:3, *v*/*v*) with different proportion of acidic modifier; flow rate, 0.5 mL/min; column temperature, 20 °C.

**Figure 4 molecules-26-07288-f004:**
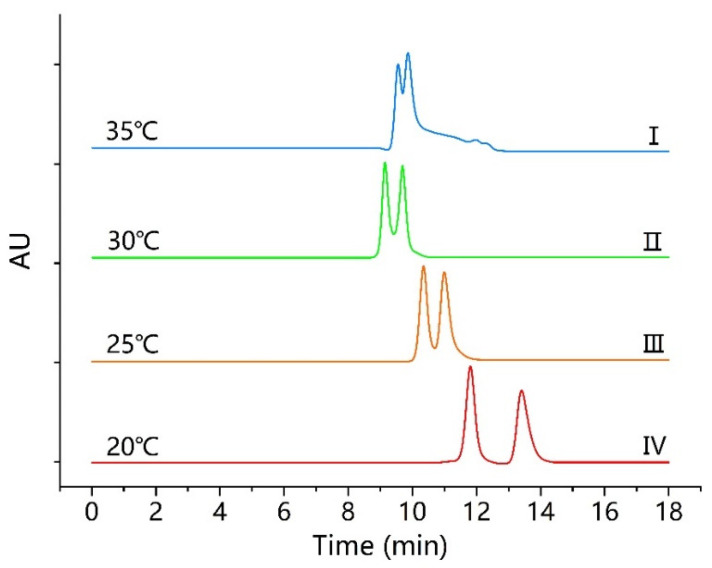
Chromatograms of closantel enantioseparation at different temperatures on Poroshell 120 CF6. Chromatographic condition: mobile phase, *n*-hexane-IPA-TFA (97:3:0.1, *v*/*v*/*v*); flow rate, 0.5 mL/min.

**Figure 5 molecules-26-07288-f005:**
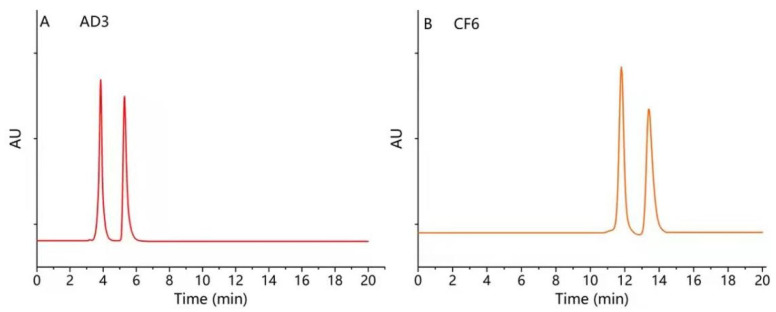
The typical chromatograms were obtained under optimized conditions on Chiralpak AD−3 (**A**) and Poroshell 120 CF6 (**B**). (**A**) Mobile phase, *n*-hexane-IPA-TFA (55:45:0.1, *v*/*v*/*v*); flow rate, 1 mL/min; column temperature, 25 °C, (**B**) mobile phase, *n*-hexane-IPA-TFA (97:3:0.1, *v*/*v*/*v*); flow rate, 0.5 mL/min; column temperature, 20 °C.

**Figure 6 molecules-26-07288-f006:**
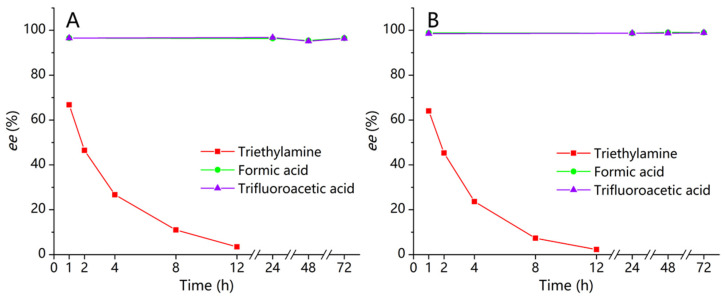
Acid-base stability of first (**A**) and second (**B**) closantel enantiomers in methanol solution within 72 h.

**Table 1 molecules-26-07288-t001:** Information of the four chiral columns (4.6 mm i.d.) used in this work.

Trade Name	Supplier	Length (cm)	Particle Size (µm)	Chiral Selectors
Enantiopak^®^ Y3R	Guangzhou Research & Creativity Biotechnology Co., Ltd. (Guangzhou, China)	25	5	Amylose[(S)-α-methyl phenyl carbamate]
Enantiopak^®^ SCDP	Guangzhou Research & Creativity Biotechnology Co., Ltd. (Guangzhou, China)	25	5	Chlorophenyl carbamoylated
Poroshell 120 CF6	Agilent Technologies, (Canada, USA)	10	2.7	isopropyl cyclofructan
Chiralpak^®^ AD−3	Diacel^®^ Corporation (Tokyo, Japan)	25	3	Amylose-3,5-dimethyl phenyl carbamate

**Table 2 molecules-26-07288-t002:** Separation results of closantel with different chiral stationary phases and alcoholic modifiers.

CSP	Alcohol Modifier	Proportion (%)	Chromatographic Separation Parameters							
			*t* _1_	*t* _2_	*k* _1_	*k* _2_	*α*	*R*s	*N* _1_	*N* _2_
Chiralpak AD−3	Ethanol	25	6.60		1.20		1.00	0		
		30	6.10	6.70	1.03	1.23	1.19	1.09	2290	2030
		35	5.58	6.50	0.86	1.17	1.36	1.26	1584	938
		40	5.2	6.00	0.73	1.00	1.47	1.35	1463	1381
		45	4.80	5.70	0.60	0.90	1.50	1.43	1246	1020
	Methanol	25	6.62	10.05	1.20	2.35	1.95	2.27	495	480
		30	6.58	10.00	1.19	2.33	1.96	2.29	519	475
		35	6.55	9.95	1.18	2.32	1.97	2.33	545	488
		40	6.50	9.90	1.16	2.30	1.98	2.36	554	492
		45	6.43	9.85	1.14	2.28	2.00	2.38	556	492
	Isopropanol	25	5.30	8.20	0.76	1.73	2.58	3.11	973	760
		30	4.50	7.00	0.50	1.31	2.62	3.24	1030	807
		35	4.20	6.0	0.40	1.06	2.65	3.37	1446	1100
		40	4.00	5.65	0.33	0.88	2.67	3.60	2010	1624
		45	3.80	5.10	0.26	0.73	2.81	3.79	3330	1845
Poroshell 120 CF6	Ethanol	20	4.15		0.38		1.00			
		15	4.55		0.51		1.00			
		10	5.00		0.66		1.00			
		5	5.52	5.75	0.84	0.91	1.09	0.44	1875	1734
		3	4.80	5.70	0.60	0.90	1.50	1.56	1888	1020
	Methanol	20	4.00		0.33		1.00	0		
		15	4.10		0.36		1.00	0		
		10	4.20		0.40		1.00	0		
		5	4.35		0.45		1.00	0		
		3	4.42		0.47		1.00	0		
	Isopropanol	20	6.30		1.10		1.00	0		
		15	6.40	6.75	1.13	1.25	1.10	1.18	7852	7705
		10	6.75	7.40	1.25	1.47	1.17	1.27	5215	2101
		5	8.70	9.75	1.90	2.25	1.18	1.77	6203	2720
		3	10.50	11.95	2.50	2.99	1.20	2.48	9773	4086
Enantiopak SCDP	Ethanol	10	18.50		5.20		1.00	0		
		30	17.90		4.97		1.00	0		
		50	17.25	18.00	4.75	5.00	1.05	0.60		
		70	16.10		4.37		1.00	0		
	Isopropanol	10	16.75		4.58		1.00	0		
		30	15.60		4.20		1.00	0		
		50	14.50	16.00	3.83	4.33	1.13	0.90		
		70	13.35		3.12		1.00	0		
Enantiopak Y3-R	Ethanol	10	15.35		4.11		1.00	0		
		30	15.00		4.00		1.00	0		
		50	14.70		3.90		1.00	0		
		70	14.10		3.70		1.00	0		
	Isopropanol	10	14.35		3.78		1.00	0		
		30	13.45		3.83		1.00	0		
		50	12.60		3.20		1.00	0		
		70	11.50		2.83		1.00	0		

*k*, retention factor; *α*, selectivity; *R*s, resolution; *N*, theoretical plate number. Mobile phase, n-hexane-alcohol −0.1% TFA; column temperature, 25 °C; flow rate, 1 mL/min for Chiralpak AD−3, Poroshell 120 CF6, Enantiopak SCDP, and Y3-R, except for Poroshell 120 CF6 flow rate, 0.5 mL/min.

**Table 3 molecules-26-07288-t003:** The chromatographic separation parameters of closantel enantiomers in Chiralpak AD−3 and Poroshell 120 CF6 columns with different acidic modifiers.

CSP	Mobile Phase	Type of Acidic Modifiers	The Proportion of Acidic Modifiers	Chromatographic Separation Parameter			
				*α*	*R*s	*N* _1_	*N* _2_
Chiralpak AD−3	*n*-hexane/isopropanol (55:45, *v*/*v*)	Trifluoroacetic acid	0.05	2.81	3.75	3330	1845
			0.10	2.81	3.75	3330	1845
			0.15	2.81	3.75	3330	1845
			0.20	2.81	3.75	3330	1845
			0.25	2.81	3.75	3330	1845
		Acetic acid	0.10	2.27	2.80	1341	1324
			0.15	2.34	2.92	1467	1444
			0.25	2.40	3.00	1617	1584
			0.50	2.50	3.28	1831	1636
		Formic acid	0.10	2.40	2.48	836	746
			0.15	2.52	2.64	919	775
			0.25	2.64	2.81	1017	806
			0.50	2.77	2.99	1006	845
Poroshell 120 CF6	*n*-hexane/isopropanol (97:3, *v*/*v*)	Trifluoroacetic acid	0.05	1.20	1.88	4535	2528
			0.10	1.25	2.69	8864	4485
			0.15	1.24	2.45	7352	2368
			0.20	1.21	1.76	4430	1290
			0.25	1.14	1.04	3798	1181
		Acetic acid	0.10	1.11	1.18	1840	1697
			0.15	1.12	1.48	2196	1872
			0.25	1.14	1.89	2216	1310
			0.50	1.13	1.58	2123	1309
		Formic acid	0.10	1.12	1.35	2513	1994
			0.15	1.13	1.63	3343	2954
			0.25	1.14	1.86	6228	4986
			0.50	1.16	1.80	5673	3463

Mobile phase: n-hexane-isopropanoldifferent proportion of the acidic modifier; flow rate, 1 and 0.5 mL/min; column temperature, 25 °C and 20 °C for Chiralpak AD−3 and Poroshell 120 CF6, respectively.

## Data Availability

The study did not report any data.

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
