# Peer review of "Analytical Separation of Closantel Enantiomers by HPLC"

_molecules, 2021, doi:10.3390/molecules26237288_

Round 1
Reviewer 1 Report
The paper reports a study focused on the enantioseparation of closantel on several HPLC columns. Among them only the Poroshell 120 CF6 and Chiralpak AD3 were able to separate the enantiomers of closantel. The problem of separation by chiral HPLC is not often reported by current literature and this could be a positive remark to the content of this submission. However, a paper discussing the enatioeseparation of closantel should be included in Introduction and the list of references (Chromatographia, 2019, 82, 221–233). The negative observation is however related to the lack of pure enantiomers such that to identify the elution order of the two enantiomers, which is possible not to be the same on different tested HPLC columns. These questions should be addressed an answer in a new version of the manuscript, which also should take into consideration the following observations:
- The statement <The substituents attached with the derivatized group can disrupt the internal hydrogen bonds and make the analyte molecules opened and unfold> (line 170,171) is unclear and should be reformulated; what is the meaning of analyte molecule opened and unfold ?
- ADMPC enantioselectivity should be defined or the term corrected (line 160);
- Chromatograms depicted in Fig. 4 should have indications of temperature instead of letters (A, B, C, and D) or these letters should be explained in its caption; on the other hand a chromatogram from this figure should be identical with that given in Fig. 1.
- The choice of trifluoracetic acid as additive in mobile phase instead of formic or acetic acids should be correlated with their acid strength;
- Column characteristics should be included in section 2.1 and not in a supplementary data;
- the statement < they are coated with silica gel with different particles size> (line 82 should be changed with <they are loaded with….>;
- the enantiomeric excess estimated in section 3.4.1 should be defined as peak area ratio between the first and the second enantiomers (not identified), such that to understand the data given in Fig. 5.
Reviewer 2 Report
The research described was carefully and logically planned. The obtained results are good and presented both in the form of tables and graphs. They have also been properly interpreted and explained, with reference to the literature of the subject. Nevertheless, the article resembles a fragment of a pretty good doctoral dissertation. Some conclusions are at the school level, e.g. regarding the influence of column temperature on separation results.
I suggest the authors mention two articles on the same topic:
Chromatographia (2019) 82:221–233; https://doi.org/10.1007/s10337-018-3604-3 and Molecules 2021, 26, 4648. https://doi.org/10.3390/molecules26154648.
The description of Fig. 4 needs to be corrected: the present form suggests that the temperature varies from 20 to 35 (from A to C?). The interpretation presented in the text of the work indicates that the temperature varies from 35 to 20 (from A to C?). This mismatch is misleading the reader.
Reviewer 3 Report
This manuscript reported the method development for the separation of closantel enantiomers by HPLC-UV using chiral columns. The experiments were sufficiently described and data discussed. However, the manuscript needs major grammatic corrections. Many adjectives were used imprecisely; sentences were redundant; the choice of conjunctions was rather liberal. All of these greatly impacted the quality of the manuscript.
In addition, here are few suggestions for the structure or the content of the manuscript.
In the Method, please specify how a, k, Rs, etc. were calculated.
In Section 3.1, a graph should be provided to illustrate the structural differences and stationary phase-analyte interactions among the four columns. What are the structures of Enantiopak SCDP and Enantiopak Y3R? The other two columns were discussed in great detail, yet Y3R was not mentioned, and SCDP was only described as “no affinity to complete separation.” By the way, please explain what it meant by “no affinity to complete separation.”
It was a little confusing that the order of the four columns in Figure 1 and Table S1 was not the same. Can the authors please change and make the order consistent?
Reviewer 4 Report
This MS describe a column and mobile phase screening for the separation of closantel enantiomers by HPLC under normal phase mode.
Retention of the two enantiomers has been investigated on four columns with a variety of mobile phase compositions. However, the MS shows incomplete and sometimes confused data.
The bibliography should be extended.
In this current form the MS cannot be published. Major revisions are required.
The authors write they used both hexane and heptane, however heptane is never mentioned in the MS.
The authors write they used mixtures of hexane (or heptane) and methanol, however these solvents are immiscible. How did the authors menage to do?
I would recommend to shift Table 1 in the paragraph 3.1 “Effect of CSP structure on the enantiomeric separation of closantel” and to add the dependence of retention, selectivity, resolution on the % of polar modifiers for each column investigated. This table could then explain why the optimization of the separation method has been performed only on Chiralpak AD3 and Poroshell 120 CF6 columns.
For a better readability of the data, I would also suggest to add a figure “k vs %” for each column (in the MS or in the SI).
A figure showing the separation on the four columns obtained with the same experimental conditions would be appreciated.
Line 32: “fields such as; chemical, […]” change to “fields such as: chemical,”
line 49: “optimize such as; the composition” change to “optimize such as: the composition”
line 62: missing verb
line 66: Filling the gap […]
line 91: it is better to indicate the dead volume, instead of dead time. Or you can indicate the flow rate.
Line 94: change “by placed” with “by placing”
line 108: “Influences of acidic and basic additives” the authors should indicate which ones
line 127: under normal phase conditions
lines 134-138: I would recommend to list the figures in alphabetical order (Fig 1A, 1B, etc..)
line 160: please define ADMPC
Lines 147-163: the authors in this discussion do not consider the amylose-based Enantiopak Y3R column. No explanation about the influence of the chiral selector on the absence of enantioresolution is proposed. The authors should discuss the reasons why retention and enantioresolution are so different on the two columns with the same backbone. They should explain possible differences in the adsorption process on the two types of chiral selectors.
Line 172-175: the authors should indicate some references or studies related to these findings.
“[...] the blocking of the inner cavity of cyclodextrin by the mobile phase components”: Were excess adsorption isotherms measured on the Enantiopak SCDP column?
Line 189-191: “In consideration of the results obtained [...]” no results have been shown yet. The authors should firstly show some results and then write conclusions.
Lines 228-232: the same result should be observed also for Enantiopak Y3R, having the same amylose polymeric backbone, is it correct?
Line 271: Figure 3B. shows
lines 288-289: the authors should explain better the sentence “as well as with the basic groups probably present on functionalized silica”
line 292-294: the authors should explain the reason why the effect of the temperature was investigated only for the Poroshell 120 CF6 column.
Line 347: the subparagraph is named “Influence of column temperature”, however it was investigated the storage temperature. Please change the name of the subparagraph.
Figure 1: this figure shows the best results obtained. It should not be placed at the beginning of the MS.
The authors should report data in a volume scale, or better with retention factors, since they are comparing columns with different dimensions (and flow rates). Moreover, the authors should indicate the name of the columns in the plot for a better readability.
Table 1: please list the results in the same order for both columns: increasing or decreasing alcohol %. Some results do not follow the expected trend, the authors should say a few words about it.
The authors should discuss why methanol leads to almost constant data in both columns, why the efficiency is so low with methanol if compared to ethanol and ipa for Chiralpak AD3 column, what is the effect of methanol on the structure of the selector for the Poroshell column.
Tables: The authors should explain why the second eluted enantiomer shows higher efficiency with respect to the first one. This trend is not usual and needs an explanation.
Fig 4: please indicate the temperature for each plot: A) ...°C, B) ...°C, etc.
A final figure should be added showing the best separation conditions for the two selected columns.
Relevant references to be added to the MS:
- D.C. Patel, Z.S. Breitbach, M.F. Wahab, C.L. Barhate, D.W. Armstrong, Gone in seconds: praxis, performance and peculiarities of ultrafast chiral liquid chromatography with superficially porous particles, Anal. Chem. 87 (2015) 9137–9148
- M. Catani, O.H. Ismail, F. Gasparrini, M. Antonelli, L. Pasti, N. Marchetti, S. Felletti, A. Cavazzini, Recent advancements and future directions of superficially porous chiral stationary phases for ultrafast high-performance enantioseparations, Analyst 142 (2017) 555–566.
- C.L. Barhate, Z.S. Breitbach, E.C. Pinto, E.L. Regalado, C.J. Welch, D.W. Armstrong , Ultrafast separation of fluorinated and desfluorinated pharmaceuticals using highly efficient and selective chiral selectors bonded to superficially porous particles, J. Chromatogr. A 1426 (2015) 241–247
- B. Chankvetadze, Recent trends in preparation, investigation and application of polysaccharide-based chiral stationary phases for separation of enantiomers in high-performance liquid chromatography, TrAC 122 (2020) 115709
- O.H. Ismail, S. Felletti, C. De Luca, L. Pasti, N. Marchetti, V. Costa, F. Gasparrini, M. Catani, The way to ultrafast, high-throughput enantioseparations of bioactive compounds in liquid and supercritical fluid chromatography, Molecules 23 (2018) 2709
- N. Khundadze, S. Pantsulaia, C. Fanali, T. Farkas, B. Chankvetadze, On our way to sub-second separations of enantiomers in high-performance liquid chromatography, J. Chromatogr. A 1572 (2018) 37–43
- A. Cavazzini, G. Nadalini, V. Malanchin, V. Costa, F. Dondi, F. Gasparrini, Adsorption mechanisms in normal-phase chromatography. mobile-phase modifier adsorption from dilute solutions and displacement effect, Anal. Chem. 79 (2007) 3802–3809
- S. Felletti, C. De Luca, G. Lievore, T. Chenet, B. Chankvetadze, T. Farkas, A. Cavazzini, M. Catani, Shedding light on mechanisms leading to convex-upward van Deemter curves on a cellulose tris(4-chloro-3-methylphenylcarbamate)-based chiral stationary phase, Journal of Chromatography A 1630 (2020) 461532
Round 2
Reviewer 3 Report
Thanks for addressing the reviewers' comments. Please correct the minor concerns below.
Please replace Figure 3 with another one with a higher resolution.
Page 15, is the sentence "Sample Availability: Samples of the compounds ... are available from the authors" complete? Please so, please remove the three periods, if not, please correct it.
Page 2, "cyano" in the full name of the Closantel should have a bracket and no space around
Author Response
please, check the response

Reviewer 4 Report
The authors have provided almost all the suggested corrections. However, one of the main points was not satisfied. The authors did not give a plausible explanation to the higher efficiency observed for the second eluted enantiomer
The reason for the higher efficiency of the second eluted enantiomer with respect to the first one has been explained in (line 292-295) with mentioning some relevant references. “In some chromatographic conditions, the (N) values were relatively higher for the second eluted enantiomer compared to the first one because the width of the second enantiomer peak width was relatively larger than the first one, the broadening of the peak might affect the value of (N) of the second enantiomer [39, 40]”.
The authors in their response write that the second enantiomer shows larger peak width, but this translates in lower N (see eq. 4) and not higher.
In Ref. 39 it is never mentioned that the second enantiomer shows better efficiency than the first and Ref. 40 takes into account two different molecules in achiral chromatography. Therefore, these 2 references should not be cited here.
I would suggest to calculate resolution and N by using the width taken at half peak height (and not at the base of the peak). As a consequence, eq. 3 and 4 have to be rewritten in the correct form.
After that, all the discussion has to be revised, as well as Tables and Figures.
Minor points:
Line 38: “concerning safety and efficiency [2-4]. Because it is well-known” correct with “concerning safety and efficiency, because it is well-known”
lines 81-83: please correct “Below is a list of four different analytical chiral columns; Enantiopak® SCDP [21], Enantiopak® Y3R [22], Chiralpak® AD3 [23], and Poroshell 120 CF6 [24] were used and
itemized in Table 1.” to “ Four different analytical chiral columns were used and are listed in Table 1: Enantiopak® SCDP [21], Enantiopak® Y3R [22], Chiralpak® AD3 [23], and Poroshell 120 CF6 [24].”
lines 107-108: change “The mobile phase was a mixture of alkanes (n-hexane) with different proportions of the alcohol modifier; EtOH, IPA, or MeOH.” with “The mobile phase was a mixture of an alkane (n-hexane) with different proportions of an alcohol modifier: EtOH, IPA, or MeOH.”
line 140: four different CSPs: Chiralpak
lines 166-167: “as well as coated onto silica matrices 3μm particles size that shows great remarkable properties in terms of enantioselectivity” this sentence is not correct. Small particle size leads to higher kinetic performance, influencing only the efficiency and not thermodynamic parameters (such as selectivity). The authors could include in the discussion the bonding density (umol/m2) of the chiral selector, that has a direct influence on chiral recognition and enantioselectivity, or delete the sentence.
Line 198: of alcohol modifiers used […]
line 208-209: delete the sentence “the difference in the particles size that have a crucial role in the enantioselectivity.” The selectivity does not depend on particle size (if the bonding density is the same). For further information you can check the paper Journal of Chromatography A, 1579 (2018) 41–48.
line 218: at the beginning of this section, I would suggest to indicate that only columns that showed an acceptable resolution (from table 2) were further investigated.
line 221-225: rewrite “ A mixture of alkanes and different alcohol modifiers are usually applied as mobile phase systems owing to their versatility in providing enantiomeric separations. Due to the
hydrogen bonding, dipole-dipole, and π–π interactions that are essential for chiral recognition, are known to be more effective under normal phase conditions. Hence, the normal mobile phase was selected to use with the previously mentioned CSPs.” with “A mixture of alkanes and different alcohol modifiers are usually applied as mobile phase owing to their versatility in providing enantiomeric separations, due to the fact that hydrogen bonding, dipole-dipole, and π–π interactions, essential for chiral recognition, are more effective under normal phase conditions. Hence, the normal mobile phase mode was selected to be use with the previously mentioned CSPs.”.
line 228: in Table 2 the range for poroshell is 3-20%
figure 2: the authors should include also k2 in the C and D plots or change the caption “influence of the alcoholic modifiers on the resolutions and retention times of closantel first eluting enantiomer”
line 273: methanol is a PROTIC solvent.
Lines 292-296: “In some chromatographic conditions, the (N) values were relatively higher for the second eluted enantiomer compared to the first one. Because the width of the second enantiomer peak width was relatively larger than the first one, the broadening of the peak might affect the value of (N) for the second enantiomer, as mentioned in previous relevant studies [39, 40].” This sentence is not correct, as already mentioned. The authors should check Rs and N values.
Lines 319-320: As shown also in Figure 3A the Rs values, achieved on Chiralpak AD3, improved from
Author Response
please, check the response
